# Mass Spectrometry Analysis of Globotriaosylsphingosine and Its Analogues in Dried Blood Spots

**DOI:** 10.3390/ijms24043223

**Published:** 2023-02-06

**Authors:** Michel Boutin, Pamela Lavoie, Margot Beaudon, Georges Kabala Ntumba, Daniel G. Bichet, Bruno Maranda, Christiane Auray-Blais

**Affiliations:** 1Division of Medical Genetics, Department of Pediatrics, Faculty of Medicine and Health Sciences, Centre de Recherche–CIUSSS de l’Estrie-CHUS, Université de Sherbrooke, Sherbrooke, QC J1H 5N4, Canada; 2Institut de Pharmacologie, Université de Sherbrooke, Sherbrooke, QC J1H 5N4, Canada; 3Research Center, Hôpital du Sacré-Coeur de Montreal, University of Montreal and Nephrology Service, Montreal, QC H4J 1C5, Canada

**Keywords:** globotriaosylsphingosine, biomarkers, Fabry disease, dried blood spots, mass spectrometry, Capitainer^®^B, lyso-Gb_3_ analogues

## Abstract

Fabry disease (FD) is an X-linked lysosomal storage disorder where impaired α-galactosidase A enzyme activity leads to the intracellular accumulation of undegraded glycosphingolipids, including globotriaosylsphingosine (lyso-Gb_3_) and related analogues. Lyso-Gb_3_ and related analogues are useful biomarkers for screening and should be routinely monitored for longitudinal patient evaluation. In recent years, a growing interest has emerged in the analysis of FD biomarkers in dried blood spots (DBSs), considering the several advantages compared to venipuncture as a technique for collecting whole-blood specimens. The focus of this study was to devise and validate a UHPLC-MS/MS method for the analysis of lyso-Gb_3_ and related analogues in DBSs to facilitate sample collection and shipment to reference laboratories. The assay was devised in conventional DBS collection cards and in Capitainer^®^B blood collection devices using both capillary and venous blood specimens from 12 healthy controls and 20 patients affected with FD. The measured biomarker concentrations were similar in capillary and venous blood specimens. The hematocrit (Hct) did not affect the correlation between plasma and DBS measurements in our cohort (Hct range: 34.3–52.2%). This UHPLC-MS/MS method using DBS would facilitate high-risk screening and the follow-up and monitoring of patients affected with FD.

## 1. Introduction 

Fabry disease (OMIM No. 301500) is an X-linked heterogenous lysosomal storage disease caused by pathogenic mutations in the *GLA* gene [1]. These mutations lead to impaired α-galactosidase A (EC 3.2.1.22) enzyme activity and to the subsequent intracellular accumulation of undegraded glycosphingolipids, including globotriaosylceramide (Gb_3_) [2,3,4,5,6], globotriaosylsphingosine (lyso-Gb_3_) [7,8], galabiosylceramide (Ga_2_) [9,10], and related isoforms and analogues. The clinical manifestations are progressive and multisystemic. The signs and symptoms in children and adolescents are mainly angiokeratomas, neuropathic pain/acroparesthesia, hypohidrosis/anhidrosis, corneal changes, proteinuria, gastrointestinal problems, and fatigue [11]. Adults are prone to major organ involvement, including an impaired renal function, cardiomyopathy, and stroke [12]. Although Fabry disease is of X-linked inheritance, women are also affected, sometimes as severely as men [13]. Since 2001, enzyme replacement therapy (ERT) is available for the treatment of Fabry disease when the indications for treatment are met, either with agalsidase-alfa (0.2 mg/kg/2 weeks) or agalsidase-beta (1.0 mg/kg/2 weeks) [12]. Migalastat, a pharmacological chaperone therapy, has also been approved (123 mg/2 days) for the treatment of Fabry disease in patients with amenable mutations [14,15]. Future therapies are currently being investigated, including second-generation ERT, substrate reduction therapy, mRNA therapy, and gene therapy [16,17,18].

Our group previously discovered the presence of molecules structurally related to lyso-Gb_3_ in urine and plasma specimens obtained from Fabry patients as part of metabolomic studies using time-of-flight mass spectrometry. The lyso-Gb_3_ analogues have modifications on the lyso-Gb_3_ sphingosine moiety that were confirmed by exact mass measurements: −C_2_H_4_ (−28 Da), −C_2_H_4_+O (−12 Da), −H_2_ (−2 Da), −H_2_+O (+14 Da), +O (+16 Da), +H_2_O (+18 Da), +H_2_O_2_ (+34 Da), and +H_2_O_3_ (+50 Da). Lyso-Gb_3_ and related analogues have been found to be useful biomarkers for screening Fabry patients [19,20] and for evaluating their response to treatment. Our group previously found that levels of lyso-Gb_3_ and some related analogues had a positive association with the left ventricular mass index and Mainz Severity Score Index in a cohort of patients with the IVS4 + 919G>A late-onset cardiac variant mutation in Taiwan [20]. A recent study highlighted that the serum lyso-Gb_3_ level and pretreatment exposure to lyso-Gb_3_ is a significant risk factor associated with adverse clinical outcomes [21]. Another group suggested that plasma lyso-Gb_3_ might be a useful indicator of the biochemical response to migalastat [22]. A group of experts thus recently suggested that plasma lyso-Gb_3_ should be routinely monitored in patients with classic Fabry disease at baseline, and then every 6 to 12 months after the treatment initiation or changes in the treatment regimen for a longitudinal evaluation [23]. 

In recent years, a growing interest has emerged for the analysis of lyso-Gb_3_ in dried blood spot (DBS) specimens [24,25,26,27,28]. DBS collection cards have several advantages compared to venipuncture as a technique for collecting whole-blood specimens [29,30,31]. The collection of whole blood by venipuncture requires the presence of a certified phlebotomist and is usually collected in a health care facility, while DBS collection cards can be used successfully at home by a minimally trained individual. Blood specimens collected by venipuncture must be processed rapidly by a laboratory technician to avoid sample or analyte degradation, while DBS specimens are simply allowed to dry. The risk of infection transmission is also lowered considering that the specimen is dried, and that a very low volume of whole blood is applied to the sampling card [29,31]. The low volume of whole blood needed to obtain a DBS also facilitates sample collection in pediatric patients [24], and regular mail can be used for the shipment of samples to reference laboratories, without the need for expensive temperature-regulated transport. Biomarker stability is usually good in DBSs, and specimens are easy to store in a cost-effective manner. However, DBS sampling using conventional collection cards can have some drawbacks, including risks of insufficient specimens [32] or uneven blood distribution on the filter card [29]. The hematocrit (Hct) level (or the volume percentage of red blood cells in blood) also influences the spreading of whole blood on the filter card and leads to sample amount variability when a disk with a fixed diameter is punched from the DBS [29,33]. Several next-generation volumetric microsampling devices designed to collect dried blood are now available to overcome these issues [33,34,35,36,37,38]. Delahaye et al. [37] recently published an extensive overview of commercially available devices, including: the Mitra^®^ device with volumetric absorptive microsampling, or VAMS^®^ technology (Neoteryx, Torrance, CA, USA); the hemaPEN^®^ (Trajan Scientific and Medical, Victoria, Australia); the HemaXis™ DB 10 (DBS System SA, Gland, Switzerland); the HemaSpot™ HF (Spot on Sciences, San Francisco, CA, USA); the Tasso-M20 (Tasso Inc., Seattle, WA, USA); and the Capitainer^®^B (Capitainer Ab, Solna, Sweden) [33,34,35,36,37,38]. The Capitainer^®^B device consists of a capillary microchannel with a precise volume of 10 µL [33,38]. When a drop of blood comes into contact with the inlet port of the device, 10 µL of blood is aspirated in the capillary microchannel, and then a thin film dissolves and the excess blood is collected onto a waste filter paper. A similar film dissolves at the outlet afterwards, and an exact volume of 10 µL of blood is absorbed in a pre-perforated paper disk by capillary forces. This principle allows the collection of a fixed volume of blood and overcomes the hematocrit bias. 

The overall objectives of this research project were thus: (1) to develop and validate a UHPLC-MS/MS method for the analysis of lyso-Gb_3_ and related analogues in DBSs collected using two different types of collection kits: conventional Whatman-GE 903 collection cards (referred to as W-DBS) and Capitainer^®^B blood collection devices (referred to as CB-DBS); (2) to establish reference values; (3) to compare the biomarker measurements obtained in plasma and in DBSs; (4) to compare the biomarker measurements in DBSs obtained from venous blood collected by venipuncture (referred to as DBS_V_) compared to capillary blood obtained by a finger prick (referred to as DBS_C_); and (5) to evaluate if the difference between the plasma and DBS biomarker measurements is associated with Hct levels.

## 2. Results

### 2.1. Method Validation 

#### 2.1.1. Accuracy and Precision

The intra-day accuracy and precision (*n* = 5) was evaluated using spiked quality controls (sQC) at three concentration levels. The results are shown in Table 1. The accuracy was acceptable for W-DBS with mean biases of 6.2% (range: 0.9 to 14.3%), 9.6% (range: 5.1 to 16.5%), and 6.7% (range: 4.6 to 12.6%) at low (L), medium (M), and high (H) lyso-Gb_3_-^13^C_6_ concentrations, respectively. The accuracy was also acceptable for CB-DBS with mean biases of 0.1% (range: −6.5 to 6.5%), 12.3% (range: 0.7 to 35.7%), and 4.8% (range: 1.6 to 9.2%) at low, medium, and high lyso-Gb_3_-^13^C_6_ concentrations, respectively. The precision was acceptable for W-DBS with %RSDs of 4.7%, 3.9%, and 3.2% at low, medium, and high lyso-Gb_3_-^13^C_6_ concentrations, respectively. The precision was also acceptable for CB-DBS with %RSDs of 5.5%, 12.3%, and 2.9% at low, medium, and high lyso-Gb_3_-^13^C_6_ concentrations, respectively.

The intra-day precision (*n* = 5) was also evaluated for this assay using DBSs obtained from Fabry patients as quality controls (pQCs) at two concentration levels. It was thus possible to evaluate the intra-day precision for lyso-Gb_3_ and five related analogues. The results are shown in Table 2. The precision was acceptable for W-DBS with %RSDs ranging from 6.9% to 12.8% in pLQC and from 2.8% to 7.8% in pHQC. The precision was also acceptable for CB-DBS with %RSDs ranging from 2.9% to 9.9% in pLQC and from 1.9% to 9.6% in pHQC.

The inter-day accuracy and precision (*n* = 5 days) was evaluated for this assay using sQCs at three concentration levels. The results are shown in Table 3. The accuracy was acceptable for W-DBS with mean biases of 3.9% (range: 0.9 to 6.2%), 6.6% (range: 3.2 to 12.0%), and 4.3% (range: −4.0 to 9.0%) at low, medium, and high lyso-Gb_3_-^13^C_6_ concentrations, respectively. The accuracy was also acceptable for CB-DBS with mean biases of 0.0% (range: −9.9 to 6.4%), 5.0% (range: −2.7 to 12.3%), and 5.1% (range: 2.6 to 8.7%) at low, medium, and high lyso-Gb_3_-^13^C_6_ concentrations, respectively. The precision was acceptable for W-DBS with %RSDs of 2.6%, 3.8%, and 5.2% at low, medium, and high lyso-Gb_3_-^13^C_6_ concentrations, respectively. The precision was also acceptable for CB-DBS with %RSDs of 6.8%, 6.9%, and 2.1% at low, medium, and high lyso-Gb_3_-^13^C_6_ concentrations, respectively.

The inter-day precision (*n* = 5 days) was also evaluated for this assay using DBSs obtained from Fabry patients as quality controls at two concentration levels. It was thus possible to evaluate the inter-day precision for lyso-Gb_3_ and five related analogues. The results are shown in Table 4. The precision was acceptable for W-DBS with %RSDs ranging from 5.4% to 11.5% in pLQC and from 6.7% to 16.4% in pHQC. The precision was also acceptable for CB-DBS with %RSDs ranging from 5.9% to 20.0% in pLQC and from 3.0% to 21.9% in pHQC.

#### 2.1.2. Limits of Detection (LODs), Limits of Quantitation (LOQs), Recovery, and Linearity

The LODs and LOQs were evaluated for lyso-Gb_3_-^13^C_6_, lyso-Gb_3_, and related analogues at −28 Da, −2 Da, +16 Da, +18 Da, and +34 Da. The results are shown in Table 5. The LODs ranged from 0.10 to 0.32 nM for W-DBS, and from 0.15 to 0.40 nM for CB-DBS. The LOQs ranged from 0.32 to 1.08 nM for W-DBS, and from 0.55 to 1.34 nM for CB-DBS. The LODs and LOQs in plasma are shown for reference purposes [19].

The recoveries from the DBS extraction, and then from the solid phase extraction (SPE) procedures were evaluated at three concentration levels (sLQC: 0.75 nM; sMQC: 75 nM; sHQC: 250 nM) and the results are shown in Table 6. Linearity was evaluated (*n* = 6 days) and the coefficients of determination (r^2^) were always ≥ 0.998 in both the calibration curves prepared using W-DBS and CB-DBS.

#### 2.1.3. Stability

The lyso-Gb_3_ stability was evaluated in DBSs at room temperature for up to 14 days, in a refrigerator (4 °C) for 38 days, in a freezer (−20 °C) for 208 days, and after five freeze/thaw cycles using DBSs prepared with a pool of blood from control individuals fortified with lyso-Gb_3_-^13^C_6_ at low, medium, and high concentrations in duplicate (Table 7). The results showed that the stability was good under these conditions, with biases ranging from −20.5% to 6.4% for W-DBS and from −8.3% to 15.3% for CB-DBS. The stability of lyso-Gb_3_ was also evaluated in the UHPLC autosampler (4 °C) up to 48 h after the first analysis. The extracts were stable after 48 h, and biases ranged from −9.1% to 13.8%. The stability of the blood pools spiked with lyso-Gb_3_-^13^C_6_ (this solution was used to prepare the sQCs) was evaluated in triplicate just after mixing and after 3 h at room temperature; the biases ranged from −3.3% to 12.9%.

The stability of lyso-Gb_3_ and five related analogues was also evaluated under the same conditions using DBSs prepared using blood from a Fabry patient at low (pLQC) and high (pHQC) concentrations in duplicate (Table 8). The results showed that the stability was good using both types of DBS collection methods for 14 days at room temperature and for 38 days in a refrigerator (4 °C), with biases ranging from −12.6% to 15.6% and from −12.7% to 13.7%, respectively. Regarding the stability in the freezer for 208 days (−20 °C), the stability was good using both types of DBS collection methods, with biases ranging from −19.8% to 7.5%, except for the lyso-Gb_3_ analogue at −28 Da, with biases ranging from −25.4% to −18.7%. The stability in DBSs was evaluated after five freeze/thaw cycles, and the biases ranged from −23.3% to 15.0% for W-DBS and from −4.8% to 30.1% for CB-DBS. Finally, the stability of the extracts left in the UHPLC autosampler (4 °C) for 48 h after the sample preparation was good for W-DBS (biases ranged from −16.9% to 8.2%). Regarding the stability of the extracts obtained from CB-DBS, the biases ranged from −3.9% to 17.4%, except for the analogues at +18 Da and +34 Da with a lower accuracy (biases ranged from 16.1% to 36.4%).

#### 2.1.4. Matrix Effect

The matrix effect was evaluated using the post-column infusion method. The signal of lyso-Gb_3_-^13^C_6_ infused at a constant flow rate was recorded during the entire chromatographic injection time of the extracts obtained from W-DBS and CB-DBS. The results shown in Figure 1A demonstrate that the signal was mostly stable across the entire chromatogram when the extracts were injected, and comparable to the signal obtained when only the resuspension solvent was injected. The multiple reaction-monitoring (MRM) chromatogram of lyso-Gb_3_ and five related analogues is shown (Figure 1B) as a reference to compare with the analyte retention times. A minor signal enhancement was observed at the retention time of the analogue at +16 Da in the extracts obtained from the CB-DBS.

### 2.2. Normal Reference Values

Normal reference values were established for lyso-Gb_3_ and five related analogues following the analysis of DBSs collected from 12 individuals not affected with Fabry disease. The normal reference values were established at the 99th percentile for W-DBS and CB-DBS for both venous and capillary blood. Table 9 shows a statistical summary of the analyte levels with the following values: mean, minimum, maximum, median, 99th percentile, sensitivity (true positive rate), and specificity (true negative rate). These values are displayed for plasma collected from the same individuals, for comparison purposes.

### 2.3. Analysis of Collected Specimens

#### 2.3.1. Comparison of Biomarker Levels between Plasma and W-DBS_C_ Matrices

A comparison of lyso-Gb_3_ and related analogue levels measured in plasma and in W-DBS_C_ specimens was performed. The Deming regression parameters are shown in Table 10. The slope measures the amount of proportional bias, while the Y-intercept represents the systematic bias between the two methods. There was no proportional bias between the plasma and W-DBS_C_ matrices, considering that the 95% confidence interval of the slope included one for lyso-Gb_3_ and its five related analogues. The examination of the Y-intercept showed a systematic bias for lyso-Gb_3_, lyso-Gb_3_ −28 Da, and lyso-Gb_3_ −2 Da, considering that the 95% confidence interval did not include 0. There was no systematic bias for lyso-Gb_3_ +16 Da, lyso-Gb_3_ +18 Da, or lyso-Gb_3_ +34 Da between the two matrices according to the Deming regression analysis.

Bland–Altman plots are shown in Figure 2. The difference between the measurements obtained in the two matrices (plasma vs. DBS) was plotted against the average measurement. The mean difference ± standard deviation (95% limits of agreement) was 2.93 ± 1.33 nmol/L (0.31 to 5.53 nmol/L) for lyso-Gb_3_, 0.17 ± 0.15 nmol/L (−0.13 to 0.47 nmol/L) for lyso-Gb_3_ −28 Da, and 0.19 ± 0.29 nmol/L (−0.37 to 0.76 nmol/L) for lyso-Gb_3_ −2 Da. Regarding lyso-Gb_3_ +16 Da, lyso-Gb_3_ +18 Da, and lyso-Gb_3_ +34 Da, the mean differences ± standard deviation (95% limits of agreement) were 0.01 ± 0.10 nmol/L (−0.19 to 0.22 nmol/L), −0.25 ± 0.42 nmol/L (−1.07 to 0.57 nmol/L), and −0.01 ± 0.21 (−0.42 to 0.40 nmol/L), respectively.

#### 2.3.2. Comparison of Biomarker Levels between Venous and Capillary Blood Matrices

A comparison of lyso-Gb_3_ and related analogue levels measured in DBSs obtained from venous and capillary blood was performed. The Deming regression parameters are shown in Table 11. There was no proportional bias between the venous and capillary blood matrices, considering that the 95% confidence interval of the slope included one for lyso-Gb_3_ and its five related analogues, except for slight deviations for lyso-Gb_3_ −28 Da (95% CI: 1.028 to 1.327) and lyso-Gb_3_ (+34) (95% CI: 1.058 to 1.097) in the DBSs obtained from Capitainer^®^B devices. The examination of the Y-intercept showed no systematic bias between the two matrices according to the Deming regression analysis, considering that the 95% confidence interval of the Y-intercept included 0 in all cases.

#### 2.3.3. Evaluation of the Hematocrit Effect

The effect of hematocrit on the correlation between the DBS and plasma biomarker measurements was evaluated using linear regression for both the Whatman-GE 903 and Capitainer^®^B collection devices. The difference between biomarker measurements in DBSs and plasma was plotted as a function of the hematocrit values. The linear regression parameters are shown in Table 12. The 95% CI of the slopes included 0 in all cases, except for lyso-Gb_3_ +16 Da in Capitainer^®^B (95% CI: −2.853 to −0.357 nM) and lyso-Gb_3_ +18 Da in Whatman-GE 903 (95% CI: 1.552 to 9.022 nM); thus, the differences between the DBS and plasma measurements did not change as a function of hematocrit in general. In all cases, the coefficients of determination (r^2^) were <0.218, and the *p*-values for testing the null hypothesis that the overall slope is zero were >0.001. There was thus no association noted between the DBS and plasma biomarker measurement differences and the hematocrit in the present cohort, where the hematocrit values ranged from 34.3 to 52.2%.

## 3. Discussion

A method was devised for the analysis of lyso-Gb_3_ and five related analogues in DBSs. The validation of the assay was performed for both Whatman-GE 903 filter paper cards and in Capitainer^®^B collection devices, and it showed acceptable intra-day and inter-day accuracy and precision. We have found that the lyso-Gb_3_ standard could not be used to prepare the calibration curve in whole blood, considering that the lyso-Gb_3_ concentration measured in healthy controls was significant. To overcome this problem, the lyso-Gb_3_-^13^C_6_ standard was chosen instead of lyso-Gb_3_ to prepare the calibration curve in whole blood, and lyso-Gb_3_-Gly was used as the internal standard. Considering that the lyso-Gb_3_ analogues have different retention times compared to the calibration and internal standards, the matrix effect was evaluated across the entire chromatographic range using the post-column infusion method. No significant ion enhancement or ion suppression regions were observed. During the method development, we found that the SPE procedure was necessary for removing a contaminant present in the Capitainer^®^B device that was causing noise during the chromatography. The LODs and LOQs were evaluated and were in the same range as those observed previously in plasma. Normal reference values were estimated in capillary and venous blood for both collection devices. There was no difference between capillary and venous blood regarding the biomarker measurements obtained according to Deming’s regression, showing that the two matrices could be used interchangeably. The sensitivity of the assay in our cohort was good, especially when looking at the whole biomarker profile, including the analogues. The sensitivity and specificity of lyso-Gb_3_ +18 Da was 100% in this cohort. The Deming regression and Bland–Altman analyses showed that there was a systematic positive bias between the measurements obtained in DBS_C_ compared to plasma for lyso-Gb_3_, lyso-Gb_3_ −28 Da, and lyso-Gb_3_ −2 Da. We hypothesized that this might be related to the presence of these molecules in red blood cells, considering that the reference values were much lower in plasma specimens from the same healthy controls. The effect of Hct was not significant in our cohort, potentially because there were no extreme Hct values (range: 34.3 to 52.2%). The main advantage of the Capitainer^®^B collection devices in this cohort was thus the sample quality, especially in a self-collection setting. There is always a risk that oversampling (multiple drops) or undersampling (insufficient specimen) would occur when using regular filter paper collection cards. In the present study, two DBSs collected with the regular filter cards were slightly insufficient, despite being collected by health care practitioners. The successful sampling indicator was also an appreciated feature on the Capitainer^®^B device and the full DBS disk was analyzed directly without the need to punch, which is convenient for the Hct effect correction. Lyso-Gb_3_ and related analogues were stable in DBSs subjected to various conditions, emphasizing the possibility of sending specimens by regular mail. The proposed assay might be useful for the high-risk screening, follow-up, and monitoring of patients affected with Fabry disease.

## 4. Materials and Methods

### 4.1. Subject Selection

A total of 20 patients affected with Fabry disease (8 males and 12 females) and 12 healthy controls (6 males and 6 females) were recruited for this study. Fabry disease was confirmed by an enzyme activity assay and/or mutation analysis. All study participants were older than 18 years of age.

### 4.2. Sample Collection and Storage

For all healthy controls and patients affected with Fabry disease, blood samples were collected by venipuncture into three 4 mL tubes containing potassium EDTA. The first tube was centrifuged at 2200 g for 10 min to retrieve plasma. The second tube was sent to the hematology laboratory to measure Hct, and the third tube was used to prepare the DBS samples. For this purpose, the blood was homogenized by gently inverting the tube 3 times. Thereafter, two blood drops of 50 µL were deposited on a Whatman-GE 903 (GE Healthcare, Chicago, IL, USA) filter paper to obtain two W-DBS_V_ samples. Two CB-DBS_V_ samples were prepared by dropping 30 µL of blood on two different ports of a Capitainer^®^B collection device (Capitainer Ab, Solna, Sweden). DBS samples were also prepared from capillary blood obtained by a fingertip puncture performed using a BD Microtainer^®^ Contact-Activated Lancet (Becton, Dickinson and Company Limited, Dublin, Ireland). Two drops of blood were deposited on a Whatman-GE 903 filter paper to obtain two W-DBS_C_ samples and two others on two different ports of a Capitainer^®^B collection device to obtain two CB-DBS_C_ samples. All DBS samples were dried for a minimum of 4 h, then inserted in a hermetic plastic bag containing a desiccant pack of 1 g (GE Healthcare, Chicago, IL, USA). The plasma and DBS samples were stored at −20 °C prior to analysis.

### 4.3. Reagents

Carbon-13-labelled globotriaosylsphingosine (lyso-Gb_3_-^13^C_6_) was purchased from GelbChem (Seattle, WA, USA). N-glycinated-globotriaosylsphingosine (lyso-Gb_3_-Gly) was synthesized in-house [39], but is also commercially available at Matreya LLC (State College, PA, USA). HPLC-grade acetonitrile (ACN) was purchased from EMD Chemicals Inc. (Darmstadt, Germany). Formic acid (FA) (99+%) was from Acros Organics (Morris Plains, NJ, USA). A.C.S.-grade o-phosphoric acid (H_3_PO_4_) (85%) and ammonium hydroxide (NH_4_OH) (29%), as well as Optima LC/MS-grade H_2_O and methanol (MeOH), were from Fisher Scientific (Fair Lawn, NJ, USA).

### 4.4. Quality Controls and Calibration Curves

Low and high DBS QCs were made with blood from two Fabry patients to evaluate the analytical precision for lyso-Gb_3_ and its analogues. Since blood always contains a small amount of lyso-Gb_3_, even in healthy individuals, lyso-Gb_3_-^13^C_6_ was spiked into a pool of freshly collected blood to prepare the QCs used to evaluate the accuracy of the method and for the calibrators. Stock solutions of lyso-Gb_3_-^13^C_6_ (in 50% ACN/50% H_2_O, 0.1% FA) were prepared to obtain concentrations 50 times higher than those targeted for the QCs and calibrators. Thereafter, the stock solutions were diluted in a ratio of 1:50 with the pool of blood and were agitated for 15 min at 22 °C on a MP980153P7 orbital shaker from VWR (Radnor, PA, USA) at a speed of 200 RPM. Finally, the W-DBS and CB-DBS samples were prepared as described previously. The lyso-Gb_3_-^13^C_6_ concentrations for the low, medium, and high spiked QCs were 0.75, 75, and 250 nM, respectively, whereas the concentrations of the 8 calibrators were 0, 0.5, 1, 2, 10, 50, 200, and 400 nM. All the DBS QCs and calibrators were prepared at the same time with fresh blood and stored at −20 °C until analysis.

### 4.5. Sample Preparation

For each study participant, a 5 mm disc was punched from a W-DBS, and a whole DBS was removed using forceps from the Capitainer^®^B collection device. Each filter paper disc was deposited into a 2 mL polypropylene tube and extracted according to a method adapted from Polo et al. [40]. Briefly, 200 µL of extraction solution (80% MeOH/15% ACN/5% H_2_O) containing 1 nM of lyso-Gb_3_-Gly used as the internal standard (IS) was added to each tube. The samples were incubated for 60 min at 45 °C in a MP980153P7 orbital shaker from VWR (Radnor, PA, USA) at a speed of 500 RPM. Afterward, 100 µL of water and 325 µL of MeOH were added to each tube. After shaking, the solution was transferred to a glass tube and the paper disc was discarded. A volume of 500 µL of H_3_PO_4_ 2% in water was added to each sample prior to its SPE purification using a mixed-mode strong cation exchange (MCX) cartridge (Oasis, 1 cc, 30 mg, LP) (Waters Corp., Milford, MA, USA) according to a protocol previously developed for the analysis of lyso-Gb_3_ and its analogues in plasma [19]. Briefly, the samples were loaded onto cartridges previously conditioned with 1 mL of MeOH and 1 mL of H_3_PO_4_ 2%. The cartridges were then washed with 1 mL of 2% FA in water followed by 1 mL of 0.2% FA in MeOH, and the samples were eluted with 600 µL of 2% NH_4_OH in MeOH. Finally, the samples were evaporated under a nitrogen stream and 100 µL of the resuspension solution (50% ACN/50% H_2_O + 0.1% FA) was added prior to analysis by UHPLC-MS/MS.

### 4.6. UHPLC-MS/MS Analysis

Lyso-Gb_3_ and its analogues were analyzed in plasma by UHPLC-MS/MS according to a previously published procedure [19], which was then adapted for the analysis of the same molecules in DBS samples. For the DBS samples, the injection volume was increased from 7.5 to 10 µL and lyso-Gb_3_-^13^C_6_ was used for the calibration curve instead of lyso-Gb_3_. Unfortunately, it was not possible to analyze the lyso-Gb_3_ analogue +50 Da in DBS due to a lack of sensitivity and reproducibility. The analyses were performed on a Acquity I-Class ultra-performance liquid chromatography system (Waters Corp., Milford, MA, USA) coupled to a Xevo TQ-S tandem mass spectrometer (Waters Corp., Milford, MA, USA) operated in the multiple reaction monitoring (MRM) mode. The UHPLC-MS/MS results were analyzed with the TargetLynx V4.2 (SCN982) software (Waters Corp., Milford, MA, USA). The calibration curve was quadratic, the origin was excluded, and a 1/x weighing factor was applied.

### 4.7. Method Validation

Method validation was performed in parallel using the W-DBS and CB-DBS QCs. The intra-day (*n* = 5) and inter-day precision (*n* = 5), the stability at −20, 4 and 22 °C (*n* = 2), the stability of prepared samples at 4 °C in the autosampler (*n* = 2), and the stability after five freeze/thaw cycles (*n* = 2) were evaluated using the patient (p) and spiked (s) QCs. A freeze/thaw cycle consisted of leaving the sample for 1 hr in the freezer followed by one hour at 22 °C. It is noteworthy to mention that the pQCs used as part of the method validation were collected using a previous model of the Capitainer^®^B collection device with a sampling volume of 13.5 µL instead of 10 µL. The intra-day (*n* = 5) and inter-day (*n* = 5) accuracy was measured for lyso-Gb_3_ using the sQCs. For each analyte, the LODs and LOQs were defined as 3 and 10 times the background noise, respectively. The adhesion to plastic and glassware was previously evaluated [19]. To measure the lyso-Gb_3_ extraction recovery, the three sQCs (*n* = 5) and control samples (DBS without lyso-Gb_3_-^13^C_6_) (*n* = 15) were extracted without IS. Just before the SPE, the control samples were spiked with the amount of lyso-Gb_3_-^13^C_6_ contained in the different QCs. The IS was also spiked in all the samples (sQCs and control samples) just before the SPE procedure. Then, the samples were processed as described earlier and the lyso-Gb_3_-^13^C_6_ concentrations measured in the sQCs were compared to the ones obtained for the spiked controls. A similar approach was used to evaluate the recovery from the SPE procedure. This time, 30 control samples were extracted without IS. For each of the three sQC concentrations, 5 control samples were spiked before the SPE and 5 others were spiked after. For all the samples, the IS was added after the SPE procedure. The recovery was obtained by comparing the lyso-Gb_3_-^13^C_6_ concentrations in the samples spiked before the SPE with the ones spiked after. The stability of the blood pools spiked with lyso-Gb_3_-^13^C_6_ and used as sQCs was tested by preparing DBS samples just after their mixture (*n* = 3) and 3 h later (*n* = 3). The matrix effect was evaluated by post-column infusion. Lyso-Gb_3_-^13^C_6_ (500 nM in 50% ACN/50% H_2_O + 0.1% FA) was infused at a flow rate of 10 µL/minute and combined with the effluent of the UHPLC system during the analysis of a blank (reconstitution solution) and control W-DBS and CB-DBS samples.

### 4.8. Statistical Analyses

Statistical analyses were performed using GraphPad Prism v9.4.1. (Graphpad Software, San Diego, CA, USA). The Deming regression and a Bland–Altman analysis were used to compare biomarker measurements obtained in plasma and W-DBS_C_. The difference between W-DBS_C_ and plasma measurements was plotted against the mean biomarker measurements. The Deming regression was used to compare the biomarker measurements obtained in capillary and venous blood in both W-DBS and CB-DBS. The effect of Hct on the difference between the biomarker measurements obtained in DBSs and plasma was evaluated using linear regression. The sensitivity of the assay was calculated as follows: TP/(TP+FN), where TP = true positives and FN = false negatives, while the specificity was calculated as follows: TN/(TN+FP), where TN = true negatives and FP = false positives. The normality of the distributions was assessed using the Shapiro–Wilk test and outliers were identified using Grubbs’s test. The statistical significance was set at *p* < 0.001.

## Figures and Tables

**Figure 1 ijms-24-03223-f001:**
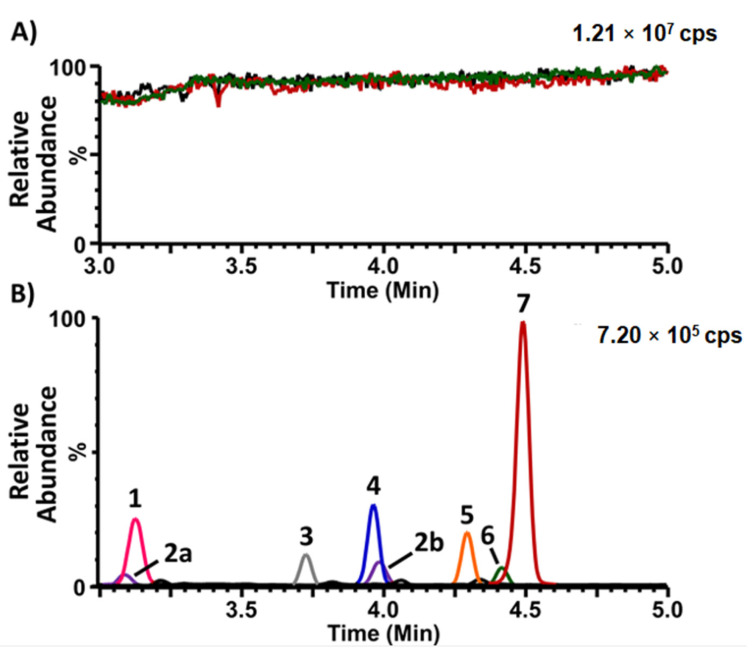
Matrix effect evaluation for the analysis of lyso-Gb_3_ and its analogues in dried blood spots (DBSs). (**A**). Signal obtained from the post-column infusion of lyso-Gb_3_-^13^C_6_ (500 nM in: 50% acetonitrile, 49.9% H_2_O, 0.1% formic acid) at a flow rate of 10 µL/min during the UHPLC injection of: (1) the resuspension solvent (50% acetonitrile, 49.9% H_2_O, 0.1% formic acid) (green line); (2) a Whatman-GE 903-collected DBS prepared with control blood (post-sample preparation, red line); and (3) a Capitainer^®^B-collected DBS prepared with control blood (post-sample preparation, black line). (**B**). Multiple reaction-monitoring (MRM) analysis of lyso-Gb_3_ and its analogues in a Whatman-GE 903-collected DBS prepared with blood from an untreated Fabry male; 1: analogue (+16 Da); 2a and 2b: (analogue (+34 Da); 3: analogue (−28 Da); 4: analogue (−2 Da); 5: lyso-Gb_3_-Gly (internal standard); 6: analogue (+18 Da); and 7: lyso-Gb_3_). cps = counts per second.

**Figure 2 ijms-24-03223-f002:**
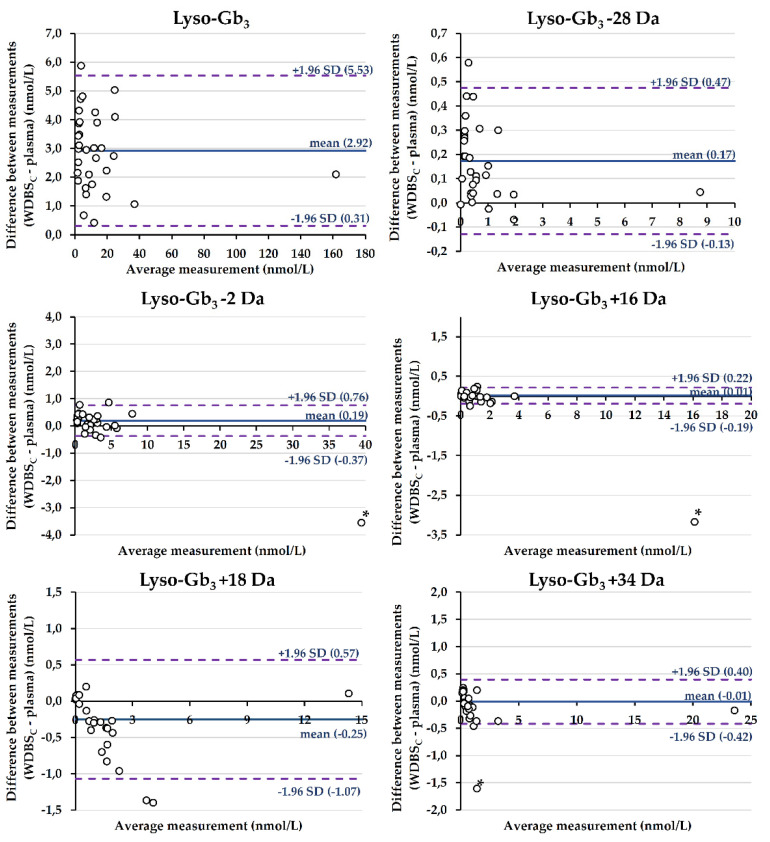
Bland–Altman analyses between lyso-Gb_3_ and related analogue measurements in plasma and in DBSs collected using Whatman-GE 903 filter papers (capillary blood) (*n* = 32 participants). Mean differences are shown by solid lines, while the dashed lines represent the limits of agreement. For lyso-Gb_3_ −2 Da, lyso-Gb_3_ +16 Da, and lyso-Gb_3_ +34 Da, an outlier (*) was excluded from the calculations of the mean difference and 95% limits of agreement.

**Table 1 ijms-24-03223-t001:** Intra-day assays (*n* = 5) performed with spiked quality control (sQC) dried blood spots (DBSs) prepared at three concentration levels by spiking lyso-Gb_3_-^13^C_6_ in a pool of blood from control individuals.

	Whatman-GE 903						Capitainer^®^B					
	sLQC		sMQC		sHQC		sLQC		sMQC		sHQC	
	nM	Bias%	nM	Bias%	nM	Bias%	nM	Bias%	nM	Bias%	nM	Bias%
Spiked concentration	0.75	na	75.00	na	250.00	na	0.75	na	75.00	na	250.00	na
Replicate 1	0.79	4.7	82.93	10.6	261.79	4.7	0.79	5.2	75.56	0.7	272.94	9.2
Replicate 2	0.80	6.5	87.37	16.5	262.81	5.1	0.73	−2.9	101.79	35.7	261.68	4.7
Replicate 3	0.76	0.9	80.88	7.8	261.57	4.6	0.80	6.5	79.88	6.5	255.61	2.2
Replicate 4	0.78	4.4	78.81	5.1	266.48	6.6	0.70	−6.5	84.77	13.0	265.91	6.4
Replicate 5	0.86	14.3	81.09	8.1	281.55	12.6	0.74	−1.6	79.18	5.6	254.11	1.6
Mean	0.80	6.2	82.21	9.6	266.84	6.7	0.75	0.1	84.23	12.3	262.05	4.8
SD	0.04	na	3.23	na	8.46	na	0.04	na	10.35	na	7.71	na
RSD%	4.67	na	3.93	na	3.17	na	5.55	na	12.28	na	2.94	na
Bias% min	na	0.9	na	5.1	na	4.6	na	−6.5	na	0.7	na	1.6
Bias% max	na	14.3	na	16.5	na	12.6	na	6.5	na	35.7	na	9.2

na = not applicable; SD = standard deviation; RSD = relative standard deviation; min = minimum; max = maximum; sLQC = spiked low quality control (0.75 nM); sMQC = spiked medium quality control (75 nM); sHQC = spiked high quality control (250 nM).

**Table 2 ijms-24-03223-t002:** Intra-day assays (*n* = 5) performed with dried blood spots (DBSs) obtained from Fabry patients as quality controls (pQCs).

				Lyso-Gb_3_ Analogues									
		Lyso-Gb_3_		−28 Da		−2 Da		+16 Da		+18 Da		+34 Da	
		pLQC	pHQC	pLQC	pHQC	pLQC	pHQC	pLQC	pHQC	pLQC	pHQC	pLQC	pHQC
		nM	nM	nM	nM	nM	nM	nM	nM	nM	nM	nM	nM
**(A) Whatman-** **GE 903**	Replicate 1	47.93	126.95	2.24	7.34	9.27	27.09	3.33	11.68	2.73	9.89	4.34	16.67
	Replicate 2	45.75	120.97	2.38	7.10	8.52	25.63	3.81	11.43	2.42	10.11	3.99	15.84
	Replicate 3	54.67	122.34	2.99	7.38	10.62	27.25	4.20	11.74	3.12	10.03	4.90	16.19
	Replicate 4	49.73	114.15	2.33	6.83	9.16	23.91	3.65	10.41	2.64	9.49	4.49	15.04
	Replicate 5	51.98	114.80	2.80	6.47	9.93	23.29	3.85	9.81	2.78	9.59	4.79	16.02
	Mean	50.01	119.84	2.55	7.02	9.50	25.43	3.77	11.01	2.74	9.82	4.50	15.95
	SD	3.47	5.38	0.33	0.38	0.80	1.80	0.32	0.86	0.25	0.27	0.36	0.60
	RSD%	6.9	4.5	12.8	5.4	8.4	7.1	8.4	7.8	9.3	2.8	8.0	3.7
**(B) Capitainer** ^®^ **B**	Replicate 1	67.70	173.59	4.54	11.04	15.80	40.05	6.87	18.24	5.31	16.04	8.66	24.76
	Replicate 2	70.01	189.64	4.36	11.41	14.74	40.74	6.15	18.04	5.96	16.95	8.70	25.05
	Replicate 3	77.03	184.55	4.37	11.79	15.47	42.14	6.29	18.50	6.85	16.17	9.57	25.77
	Replicate 4	73.57	190.14	4.13	11.40	14.95	42.13	6.17	18.81	5.85	19.35	9.34	28.32
	Replicate 5	75.43	185.82	3.99	11.71	14.98	42.51	6.30	18.87	6.52	19.47	9.26	29.27
	Mean	72.75	184.75	4.28	11.47	15.19	41.51	6.36	18.49	6.10	17.59	9.10	26.63
	SD	3.85	6.68	0.22	0.30	0.43	1.06	0.29	0.36	0.60	1.69	0.41	2.03
	RSD%	5.3	3.6	5.1	2.6	2.9	2.6	4.6	1.9	9.9	9.6	4.5	7.6

SD = standard deviation; RSD = relative standard deviation; pLQC = low quality control prepared with blood from a Fabry patient; pHQC = high quality control prepared with blood from a Fabry patient.

**Table 3 ijms-24-03223-t003:** Inter-day (*n* = 5) assays performed with spiked quality control (sQC) dried blood spots (DBSs) prepared at three concentration levels by spiking lyso-Gb_3_-^13^C_6_ in a pool of blood from control individuals.

	Whatman-GE 903						Capitainer^®^B					
	sLQC		sMQC		sHQC		sLQC		sMQC		sHQC	
	nM	Bias%	nM	Bias%	nM	Bias%	nM	Bias%	nM	Bias%	nM	Bias%
Spiked concentration	0.75	na	75.00	na	250.00	na	0.75	na	75.00	na	250.00	na
Day 1	0.80	6.2	82.21	9.6	266.84	6.7	0.75	0.1	84.23	12.3	262.05	4.8
Day 2	0.79	5.1	84.00	12.0	272.55	9.0	0.80	6.4	73.06	−2.6	260.73	4.3
Day 3	0.76	0.9	77.69	3.6	270.24	8.1	0.80	6.3	80.39	7.2	271.74	8.7
Day 4	0.76	1.1	77.38	3.2	239.97	−4.0	0.68	−9.9	72.95	−2.7	262.98	5.2
Day 5	0.80	6.2	78.29	4.4	254.29	1.7	0.73	−2.7	82.95	10.6	256.45	2.6
Mean	0.78	3.9	79.91	6.6	260.78	4.3	0.75	0.0	78.72	5.0	262.79	5.1
SD	0.02	na	3.00	na	13.60	na	0.05	na	5.40	na	5.60	na
RSD%	2.6	na	3.8	na	5.2	na	6.8	na	6.9	na	2.1	na
Bias% minimum	na	0.9	na	3.2	na	−4.0	na	−9.9	na	−2.7	na	2.6
Bias% maximum	na	6.2	na	12.0	na	9.0	na	6.4	na	12.3	na	8.7

na = not applicable; SD = standard deviation; RSD = relative standard deviation; sLQC = spiked low quality control (0.75 nM); sMQC = spiked medium quality control (75 nM); sHQC = spiked high quality control (250 nM).

**Table 4 ijms-24-03223-t004:** Inter-day (*n* = 5) assays performed with dried blood spots (DBSs) obtained from Fabry patients as quality controls (pQCs).

				Lyso-Gb_3_ Analogues									
		Lyso-Gb_3_		−28 Da		−2 Da		+16 Da		+18 Da		+34 Da	
		pLQC	pHQC	pLQC	pHQC	pLQC	pHQC	pLQC	pHQC	pLQC	pHQC	pLQC	pHQC
		nM	nM	nM	nM	nM	nM	nM	nM	nM	nM	nM	nM
**(A) Whatman-** **GE 903**	Day 1	50.01	119.84	2.55	7.02	9.50	25.43	3.77	11.01	2.74	9.82	4.50	15.95
	Day 2	44.51	122.23	2.38	7.06	8.35	25.50	3.55	11.11	2.83	8.50	4.09	13.75
	Day 3	40.91	115.93	2.05	5.87	7.26	22.49	3.46	10.81	2.48	9.45	4.72	15.22
	Day 4	40.53	113.24	2.04	6.20	7.60	24.10	3.96	13.34	2.85	7.51	5.04	14.98
	Day 5	40.53	102.58	2.04	5.50	7.44	21.36	3.79	12.67	2.75	6.49	4.76	13.35
	Mean	43.30	114.76	2.21	6.33	8.03	23.78	3.71	11.79	2.73	8.35	4.62	14.65
	SD	4.11	7.64	0.24	0.69	0.92	1.82	0.20	1.14	0.15	1.37	0.35	1.07
	RSD%	9.5	6.7	10.7	11.0	11.5	7.7	5.4	9.7	5.4	16.4	7.6	7.3
**(B) Capitainer** ^®^ **B**	Day 1	72.75	184.75	4.28	11.47	15.19	41.51	6.36	18.49	6.10	17.59	9.10	26.63
	Day 2	66.06	194.13	3.66	10.37	13.50	38.14	5.59	15.69	5.89	19.45	8.39	26.80
	Day 3	70.66	186.87	3.70	9.66	14.02	38.99	6.70	18.68	4.22	11.66	7.22	23.71
	Day 4	75.98	197.24	3.80	9.62	15.09	38.51	7.71	19.81	4.57	13.21	7.69	22.67
	Day 5	78.69	196.27	3.43	10.29	15.54	42.21	6.94	20.63	6.87	19.08	10.07	29.24
	Mean	72.83	191.85	3.77	10.28	14.67	39.87	6.66	18.66	5.53	16.20	8.49	25.81
	SD	4.87	5.68	0.31	0.75	0.87	1.86	0.78	1.88	1.10	3.55	1.13	2.63
	RSD%	6.7	3.0	8.3	7.3	5.9	4.7	11.7	10.1	20.0	21.9	13.3	10.2

SD = standard deviation; RSD = relative standard deviation; pLQC = low quality control prepared with blood from a Fabry patient; pHQC = high quality control prepared with blood from a Fabry patient.

**Table 5 ijms-24-03223-t005:** Limits of detection (LODs) and limits of quantification (LOQs) for lyso-Gb_3_ and its analogues in dried blood spots (DBSs).

	LOD			LOQ		
	Whatman-GE 903	Capitainer^®^B	Plasma	Whatman-GE 903	Capitainer^®^B	Plasma
	nM	nM	nM	nM	nM	nM
Lyso-Gb_3_-^13^C_6_	0.21	0.40	na	0.70	1.34	na
Lyso-Gb_3_	0.32	0.37	0.23	1.08	1.23	0.77
Lyso-Gb_3_ (Analogue − 28 Da)	0.11	0.18	0.06	0.36	0.59	0.21
Lyso-Gb_3_ (Analogue −2 Da)	0.22	0.15	0.29	0.73	0.55	0.97
Lyso-Gb_3_ (Analogue +16 Da)	0.10	0.39	0.22	0.32	1.29	0.72
Lyso-Gb_3_ (Analogue +18 Da)	0.13	0.23	0.14	0.42	0.77	0.47
Lyso-Gb_3_ (Analogue +34 Da)	0.13	0.19	0.24	0.43	0.64	0.79

LOD = limit of detection; LOQ = limit of quantification; na = not applicable.

**Table 6 ijms-24-03223-t006:** Dried blood spot (DBS) and solid phase extraction (SPE) recoveries for lyso-Gb_3_-^13^C_6_ in DBSs (*n* = 5).

	DBS Extraction Recovery (%)		SPE Recovery (%)	
	Whatman-GE 903	Capitainer^®^B	Whatman-GE 903	Capitainer^®^B
sLQC	69	68	71	72
sMQC	71	66	72	69
sHQC	66	68	74	74

DBS = dried blood spot; SPE = solid phase extraction; sLQC = spiked low quality control (0.75 nM); sMQC = spiked medium quality control (75 nM); sHQC = spiked high quality control (250 nM).

**Table 7 ijms-24-03223-t007:** Stability assays (*n* = 2) performed with spiked quality control (sQC) dried blood spots (DBSs) prepared at three concentration levels by spiking lyso-Gb_3_-^13^C_6_ in a pool of blood from control individuals.

	Whatman-GE 903			Capitainer^®^B		
	sLQC	sMQC	sHQC	sLQC	sMQC	sHQC
	Bias%	Bias%	Bias%	Bias%	Bias%	Bias%
Room temperature, 22 °C (7 days)	−9.6	−1.3	−4.2	−8.3	−1.9	−4.3
Room temperature, 22 °C (14 days)	−20.5	−3.2	−3.0	−3.2	−7.8	0.7
Refrigerator, 4 °C (38 days)	−8.3	−6.9	−2.2	15.3	3.1	−3.4
Freezer, −20 °C (208 days)	−6.3	−0.3	−1.2	3.8	1.2	−1.0
Freeze/thaw cycles (*n* = 5)	−8.7	6.4	1.9	10.3	−0.2	1.2
UHPLC Autosampler, 4 °C (48 h)	13.8	−1.9	−8.4	−9.1	−2.4	−2.7

sLQC = spiked low quality control (0.75 nM); sMQC = spiked medium quality control (75 nM); sHQC = spiked high quality control (250 nM).

**Table 8 ijms-24-03223-t008:** Stability assays (*n* = 2) performed with dried blood spots (DBSs) from Fabry patients as quality controls (pQCs).

				Lyso-Gb_3_ Analogues									
		Lyso-Gb_3_		−28 Da		−2 Da		+16 Da		+18 Da		+34 Da	
		pLQC	pHQC	pLQC	pHQC	pLQC	pHQC	pLQC	pHQC	pLQC	pHQC	pLQC	pHQC
		Bias%	Bias%	Bias%	Bias%	Bias%	Bias%	Bias%	Bias%	Bias%	Bias%	Bias%	Bias%
**(A) Whatman-** **GE 903**	RT, 22 °C (7 days)	1.9	−4.1	2.9	−4.1	4.6	3.3	−1.1	3.7	10.2	−2.7	0.8	−0.5
	RT, 22 °C (14 days)	−9.2	4.2	−10.3	2.6	−10.4	5.7	−12.6	3.1	11.4	15.6	−6.6	5.5
	Refrigerator, 4 °C (38 days)	−5.0	10.8	2.2	12.0	−1.1	10.9	10.5	7.5	−12.7	12.0	−3.8	13.7
	Freezer, −20 °C (208 days)	−4.9	−4.9	−25.4	−19.6	−19.8	−7.9	−14.8	−9.6	7.4	−15.4	7.5	−4.0
	Freeze/thaw cycles (*n* = 5)	−16.0	6.8	−18.9	5.2	−17.5	−1.4	−23.4	0.9	−7.6	15.0	−19.6	5.2
	UHPLC Autosampler, 4 °C (24 h)	−2.0	−2.1	2.3	−4.3	−1.6	−3.7	−4.2	−2.0	27.3	16.9	18.1	12.9
	UHPLC Autosampler, 4 °C (48 h)	−9.4	−16.9	−13.8	−14.4	−14.8	−16.7	−13.1	−14.4	8.2	−1.7	2.6	−2.6
**(B) Capitainer** ^®^ **B**	RT, 22 °C (7 days)	−4.2	0.8	−0.4	3.6	−0.2	−0.6	−1.7	6.6	−7.1	4.5	2.3	−0.7
	RT, 22 °C (14 days)	−2.7	−5.2	1.6	−6.6	0.3	−2.3	−8.2	−10.0	7.0	−0.9	6.6	−0.5
	Refrigerator, 4 °C (38 days)	−6.0	−3.0	−8.5	−2.8	−8.4	−1.2	1.0	−7.0	−3.1	1.6	−9.1	−5.1
	Freezer, −20 °C (208 days)	−1.3	0.0	−18.7	−19.5	−1.6	−5.5	−7.1	−12.7	−7.7	1.0	−4.8	−6.2
	Freeze/thaw cycles (*n* = 5)	−1.3	0.6	14.7	−4.0	6.7	0.7	2.1	−4.8	1.9	30.1	9.1	19.8
	UHPLC Autosampler, 4 °C (24 h)	0.2	1.1	4.5	−0.1	−2.2	−1.3	0.1	0.2	11.0	15.5	23.3	16.7
	UHPLC Autosampler, 4 °C (48 h)	−3.9	−1.8	17.4	9.8	10.8	7.8	14.0	10.0	16.1	25.9	36.4	32.4

pLQC = low quality control prepared with blood from a Fabry patient; pHQC = high quality control prepared with blood from a Fabry patient; RT = room temperature.

**Table 9 ijms-24-03223-t009:** Normal reference values (99th percentile) established for each analyte under study following the analysis of DBSs from 12 healthy reference controls. Mean, minimum, maximum, and median values are shown, along with the sensitivity and specificity of the biomarkers.

Biomarker	Blood Type	Collection Method	Mean	Min	Max	Median	99th Percentile	Sensitivity (%)	Specificity (%)
Lyso-Gb_3_ (nM)	Venous	Whatman 903	3.96	2.25	6.25	3.86	6.14	100	92
		Capitainer^®^B	4.60	3.16	8.43	4.18	8.19	85	92
	Capillary	Whatman 903	4.26	2.74	6.71	4.20	6.61	95	92
		Capitainer^®^B	4.15	2.95	5.93	4.19	5.92	100	100
	Plasma	EDTA K2	0.74	0.33	1.09	0.78	1.08	100	92
Lyso-Gb_3_ −28 Da (nM)	Venous	Whatman 903	0.25	nd	0.53	0.26	0.51	55	92
		Capitainer^®^B	nd	nd	0.42	nd	0.40	85	92
	Capillary	Whatman 903	0.28	nd	0.58	0.27	0.56	60	92
		Capitainer^®^B	nd	nd	0.24	nd	0.24	100	92
	Plasma	EDTA K2	nd	nd	nd	nd	nd	100	92
Lyso-Gb_3_ −2 Da (nM)	Venous	Whatman 903	0.49	0.33	0.79	0.44	0.78	100	92
		Capitainer^®^B	0.60	0.41	1.16	0.53	1.11	95	92
	Capillary	Whatman 903	0.56	0.35	1.03	0.53	1.00	100	92
		Capitainer^®^B	0.55	0.36	0.82	0.52	0.80	100	92
	Plasma	EDTA K2	nd	nd	0.31	nd	0.30	100	92
Lyso-Gb_3_ +16 Da (nM)	Venous	Whatman 903	nd	nd	0.21	nd	0.20	95	92
		Capitainer^®^B	nd	nd	nd	nd	nd	80	100
	Capillary	Whatman 903	nd	nd	nd	nd	nd	100	100
		Capitainer^®^B	nd	nd	nd	nd	nd	84	100
	Plasma	EDTA K2	nd	nd	nd	nd	nd	95	100
Lyso-Gb_3_ +18 Da (nM)	Venous	Whatman 903	nd	nd	nd	nd	nd	100	100
		Capitainer^®^B	nd	nd	nd	nd	nd	100	100
	Capillary	Whatman 903	nd	nd	nd	nd	nd	100	100
		Capitainer^®^B	nd	nd	nd	nd	nd	100	100
	Plasma	EDTA K2	nd	nd	nd	nd	nd	100	100
Lyso-Gb_3_ +34 Da (nM)	Venous	Whatman 903	0.28	0.17	0.39	0.25	0.38	90	92
		Capitainer^®^B	nd	nd	0.25	nd	0.25	90	92
	Capillary	Whatman 903	0.30	0.21	0.35	0.30	0.35	85	100
		Capitainer^®^B	nd	nd	0.20	nd	0.20	95	92
	Plasma	EDTA K2	nd	nd	nd	nd	nd	100	92

nd = not detected (under the limits of detection).

**Table 10 ijms-24-03223-t010:** Deming regression analysis parameters for lyso-Gb_3_ and analogue measurements in plasma and in DBSs collected using Whatman-GE 903 filter papers (capillary blood) (*n* = 32 participants).

Biomarker	Slope	95% CI (Slope)	Y-Intercept (nM)	95% CI (Y-Intercept)	*n*
Lyso-Gb_3_	0.993	0.956–1.029	3.019	2.434–3.604	32
Lyso-Gb_3_ −28 Da	0.973	0.800–1.147	0.192	0.101–0.283	32
Lyso-Gb_3_ −2 Da	0.910	0.775–1.046	0.365	0.085–0.646	32
Lyso-Gb_3_ +16 Da	0.825	0.547–1.102	0.124	−0.057–0.306	32
Lyso-Gb_3_ +18 Da	0.962	0.421–1.504	−0.195	−0.724–0.334	32
Lyso-Gb_3_ +34 Da	0.988	0.454–1.523	−0.045	−0.402–0.312	32

CI: confidence interval.

**Table 11 ijms-24-03223-t011:** Deming regression analysis parameters for lyso-Gb_3_ and analogue measurements in venous and capillary blood.

Biomarker	Collection Method	Slope	95% CI (Slope)	Y-Intercept (nM)	95% CI (Y-Intercept)	*n*
Lyso-Gb_3_	Whatman 903	1.243	0.650–1.837	−2.576	−9.022–3.869	32
	Capitainer^®^B	1.134	0.867–1.401	−1.678	−4.638–1.282	31
Lyso-Gb_3_ −28 Da	Whatman 903	1.288	0.598–1.977	−0.153	−0.556–0.250	32
	Capitainer^®^B	1.178	1.028–1.327	−0.066	−0.156–0.023	31
Lyso-Gb_3_ −2 Da	Whatman 903	1.301	0.564–2.037	−0.643	−2.203–0.917	32
	Capitainer^®^B	1.119	0.933–1.305	−0.262	−0.670–0.147	31
Lyso-Gb_3_ +16 Da	Whatman 903	1.233	0.617–1.850	−0.183	−0.595–0.229	32
	Capitainer^®^B	1.070	0.853–1.286	−0.055	−0.209–0.099	31
Lyso-Gb_3_ +18 Da	Whatman 903	1.193	0.856–1.531	−0.170	−0.437–0.098	32
	Capitainer^®^B	0.979	0.762–1.197	0.007	−0.184–0.199	31
Lyso-Gb_3_ +34 Da	Whatman 903	1.225	0.548–1.903	−0.170	−0.595–0.255	32
	Capitainer^®^B	1.077	1.058–1.097	−0.010	−0.055–0.035	31

CI: confidence interval.

**Table 12 ijms-24-03223-t012:** Simple linear regression parameters of the difference between DBSs and plasma biomarker measurements as a function of hematocrit for lyso-Gb_3_ and related analogues.

Biomarker	Collection Method	Slope	95% CI (Slope)	Y-Intercept (nM)	95% CI (Y-Intercept)	r^2^	*p*-Value	*n*
Lyso-Gb_3_	Whatman 903	8.107	−5.019–21.230	−0.466	−5.971–5.040	0.050	0.217	32
	Capitainer^®^B	−7.438	−23.140–8.261	6.358	−0.214–12.930	0.031	0.341	31
Lyso-Gb_3_ −28 Da	Whatman 903	1.272	−0.217–2.762	−0.358	−0.983–0.266	0.092	0.091	32
	Capitainer^®^B	−0.439	−1.538–0.660	0.309	−0.149–0.767	0.023	0.420	30
Lyso-Gb_3_ −2 Da	Whatman 903	−0.070	−3.106–2.965	0.222	−1.046–1.489	0.000	0.963	31
	Capitainer^®^B	−1.937	−5.551–1.677	0.993	−0.513–2.499	0.041	0.282	30
Lyso-Gb_3_ +16 Da	Whatman 903	0.130	−0.967–1.226	−0.041	−0.499–0.417	0.002	0.811	31
	Capitainer^®^B	−1.605	−2.853–(−0.357)	0.702	0.182–1.222	0.199	0.014	30
Lyso-Gb_3_ +18 Da	Whatman 903	5.287	1.552–9.022	−2.460	−4.027–(−0.893)	0.218	0.007	32
	Capitainer^®^B	1.715	−0.403–3.832	−0.790	−1.673–0.092	0.089	0.108	30
Lyso-Gb_3_ +34 Da	Whatman 903	1.434	−0.608–3.475	−0.610	−1.468–0.247	0.066	0.162	31
	Capitainer^®^B	0.981	−0.805–2.767	−0.465	−1.210–0.279	0.043	0.270	30

CI: confidence interval.

## Data Availability

Not applicable.

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
