# Peer review of "Mass Spectrometry Analysis of Globotriaosylsphingosine and Its Analogues in Dried Blood Spots"

_ijms, 2023, doi:10.3390/ijms24043223_

Round 1
Reviewer 1 Report
The lysoGB3 and related analogues were the biomarkers of Fabry disease. The authors develop the MS/MS assay of lysoGb3 and its analogues from DBS and validate with excellent results including precision, accuracy, stability ...etc. The authors also make the comparison of biomarkers between different blood matrices. Base on this study, the DBS samples could be used to detect the biomarkers level and monitor the effect of ERT. In two blood matrices (plasma and DBS ), we can observe the different normal reference ranges and the authors hypothesized that lysoGb3 might interact with RBCs. But the significant difference in lysoGb3 levels between Whatman-903 and Capitainer B (Table. 2 and Table. 4) among Fabry patients were not consistency with the lysoGbs3 levels among healthy controls (Table. 9).
The data of matrix effect and hct effect were very useful to evaluate the stability of lysoGb3 assay by UHPLC-MS/MS.
Author Response
Please see the comments for both reviewers in the two attached documents.
Thank you for the attention given to this matter.
Christiane Auray-Blais

Reviewer 2 Report
In this work, globotriaosylsphingosine and its analogues in DBS were analyzed using LC-MS. Generally speaking, the work is relatively simple with limited innovation; However, the work was done very carefully, with detailed validation of the method. There are the following problems:
1. Can the Introduction part be divided into sections for better logic?
2. Table 7, Freeze/thaw cycles (n=5). For DBS, how?
3. Table 9, what is "99th Percentile"? "100,0" and "91,7" for sensitivity and specificity, what do they mean? Why does the data in the Table (Not only Table 9) have so many commas? Do commas make sense? Feeling causes a lot of confusion.
4. Table 10-12, Is information missing? The data is not clear. Is there any other way to display it?
5. Finally, are globotriaosylsphingosine and its analogues in DBS good biomarkers for this disease?
Author Response
Please see the responses to Reviewer 2 in the attached document. Thank you for your consideration!
Kind regards,
Christiane Auray-Blais
